# Cesarean Scar Ectopic Pregnancy—Case Series: Treatment Decision Algorithm and Success with Medical Treatment

**DOI:** 10.3390/medicina57040362

**Published:** 2021-04-08

**Authors:** Lorena Sabonet Morente, Ana I. Guzmán León, M. Pilar Espejo Reina, Jose R. Anderica Herrero, Ernesto González Mesa, Jesús S. Jiménez López

**Affiliations:** FEA Obstetric and Gynecology Hospital Regional Materno Infantil, C.P.29011 Málaga, Spain; anaguzmanleon@hotmail.com (A.I.G.L.); pilarespejoreina@gmail.com (M.P.E.R.); jranderica@hotmail.com (J.R.A.H.); egonzalezmesa@gmail.com (E.G.M.); jesuss.jimenez.sspa@juntadeandalucia.es (J.S.J.L.)

**Keywords:** cesarean scar, ectopic pregnancy

## Abstract

*Background*: Cesarean scar ectopic pregnancies are a rare form of extrauterine pregnancies, yet their incidence has increased along with the rise in the number of cesarean deliveries. As with other ectopic pregnancies, cesarean scar ectopic pregnancies pose a greater risk for maternal hemorrhage and ultimately maternal mortality. *Case presentation*: We present a series of clinical cases of cesarean scar ectopic pregnancy diagnosed by transvaginal ultrasonography. Each patient received an individualized treatment: the rate of success depended on the particular maternal condition in each case. Due to the low frequency of this entity, there are no clear protocols for its treatment and thus there are numerous options for treatment and follow-up: expectant management, medical therapy, surgical intervention, uterine artery embolization or a combined approach. Each method has different levels of success and is dependent on the surgeon’s skill and patient presentation. A transvaginal ultrasound is necessary to obtain the fine details of the gestation sac and its relation to the scar and must be followed by a meticulous abdominal scan with a full bladder. *Conclusion*: Herein, we present a rare pathological phenomenon whose frequency is on the rise, and for which transvaginal ultrasound and flow Doppler provide high diagnostic accuracy. Early diagnosis of cesarean scar ectopic pregnancies offers treatment options that may help avoid uterine rupture and bleeding, thus preserving the uterus and future fertility.

## 1. Background

The secondary rise of repeat cesarean delivery has been associated with an increase in complications of embryo implantation in a previous cesarean scar (CS), resulting in a cesarean scar ectopic pregnancy (CSEP).

Ectopic pregnancy is defined as any pregnancy that implants in a location other than the uterine endometrium. While most ectopic pregnancies occur in the fallopian tube, pregnancies can also implant in the abdomen, cervix, ovary and cornua of the uterus [1,2]. A cesarean scar ectopic pregnancy (CSEP) is a developing pregnancy implanted in the myometrium of a previous cesarean delivery scar. Cesarean scar ectopic pregnancy (CSEP) is rare and occurs in approximately one in every 2000 pregnancies of patients who have had a previous cesarean section. The rate of cesarean deliveries has shown a steady increase over the past few decades. Given this increase and the improved technology of sonographic imaging, the incidence of the detection of cesarean scar ectopic pregnancies has also shown an upward tendency [3,4].

This risk of CSEP is not necessarily affected by the number of previous cesarean sections [5,6,7,8]. The most probable mechanism that can explain scar implantation is that there is invasion of the myometrium through a microtubular tract between the cesarean section scar and the endometrial canal; damage to the decidua basalis during uterine surgery can persist in the endometrium in the form of tiny dehiscent tracts or minute wedge defects [6]. Undiagnosed CSEP may progress to uterine rupture, hemorrhage, loss of future fertility, and possibly maternal death.

In this review, our objective is to present a series of clinical cases in which the early and adequate diagnosis made it possible to individualize management and achieve a successful treatment and outcome. The interest of our article lies in the absence of clinical guidelines on the management of CSEP at present (Figure 1).

## 2. Case Report

Case One: A 35-year-old woman presented at six weeks of gestation, dated according to her last menstrual cycle, with painless vaginal bleeding. The patient had regular menses and a history of cesarean delivery 3 years prior with no other significant medical history, or history of sexually transmitted infections.

The patient’s transvaginal ultrasound was notable for a uterus anteflexed with a gestational sac of 12 mm and embryo of 4 mm with positive cardiac activity located on the anterior aspect of the lower uterine segment. The anterior aspect of the myometrium measured 5.2 mm and the lower uterine segment measured 1.5 mm. We could demonstrate color Doppler signal uptake in the area that contacts the posterior lip of the cervix and the preserved bladder. The patient’s serum quantitative beta human chorionic gonadotropin (β-hCG) was 3600 UI/L. At presentation, her vitals were within normal limits and stable.

These ultrasound findings raised the suspicion of cesarean scar ectopic pregnancy. Following appropriate counseling, the patient confirmed her desire for future fertility and, understanding the risks and benefits, she agreed to medical treatment. An initial dose of intramuscular methotrexate at one milligram per kilogram (mg/kg) was administered. At one week follow-up, she received a three multi-dose regimen over a period of five days. Three doses of folinic acid were added to the treatment. After a systemic treatment of multiple doses for one week, the patient was asymptomatic and the betahCG serum level was 5809 UI/L. An aspiration curettage was thus performed in the ultrasound operating room, affording possible transvaginal access by imprinting the sac at the cervical level. During the curettage, access to the gestational sac area was achieved by obtaining decidual material. The pathological report showed decidual remains. On follow-up the next day, the patient’s serum quantitative beta-hCG was 1371 UI/L. The following week, a transvaginal ultrasound confirmed an empty uterine cavity and endometrium measuring 5 mm; an isthmocele was identified on the cesarean scar. Serum level of betahCG was 90 UI/L. The patient was monitored weekly until normalization of betahCG occurred, i.e., within three weeks.

Case Two: A 34-year-old female whose last menstrual period occurred seven weeks prior presented to the emergency department with vaginal bleeding. The patient’s medical history was notable for two deliveries and one cesarean delivery two years prior. She denied any fever, chills, abdominal pain, nausea, vomiting, chest pain, or dizziness.

A transvaginal ultrasound was performed (Figure 2), revealing a retroflexed uterus with endomyometrium measuring 6 mm, a gestational sac of 15 mm, and an embryo of 3.6 mm with positive cardiac activity located on the cesarean scar. The bladder was not invaded, and there was normal adnexal and a small amount of free fluid in the pouch of Douglas. The patient’s serum determination of betahCG was 14,508 UI/L.

After discussion with the patient regarding the ultrasound findings, potential complications of the continuation of a cesarean scar pregnancy, and reproductive goals, the patient elected for permanent sterilization. She underwent an uncomplicated total laparoscopic hysterectomy with removal of the cesarean scar pregnancy. This is a definitive treatment for the resolution of the problem in a woman with her reproductive desire fulfilled. The risk and benefits of expectant management and the necessity of a series of follow-up visits were discussed with the patient and she decided definitive treatment because she had her reproductive desire fulfilled. Although our aim was to preserve the uterus, at the patient’s express wish, a hysterectomy was performed. During the surgery, when we separated the bladder from the uterus, the bulging of the gestational sac could be clearly identified in the scar area, with great vascularization (Figure 3). The uterus did not show any other abnormal finding. The patient was discharged on the third day following surgery without complications and was scheduled for close follow-up with obstetrics and gynecology.

Case Three: A 41-year-old woman of thirteen weeks gestation presented to the emergency department for pain accompanied by vomiting and two episodes of syncope with loss of consciousness. The patient had two prior preterm cesarean sections at 27 and 33 weeks, respectively.

Upon arrival at our emergency department, the clinical examination showed pain when pressing at the hypogastric area as well as unclear Blumberg sign. A transvaginal ultrasound was performed, revealing abundant hemoperitoneum with an embryo of 64 mm with positive cardiac activity, 5 mm from the bladder within an invasion. Due to the threat of hemoperitoneum, transvaginal ultrasound was enough for the diagnosis of massive hemoperitoneum without requiring any additional tests and the suspicion of CSEP, and a diagnostic laparoscopy was performed and revealed massive hemoperitoneum with abundant blood clot. When a rupture on the anterior aspect of the lower uterine segment with output of decidual material was found along with significant active bleeding that was difficult to control by laparoscopy, we proceeded to open surgery. A wide uterine rupture was identified at the level of the uterine segment, through which the amniotic sac protruded and was expelled. Due to profuse bleeding after expulsion of the gestation together with the placenta and integral membranes, a subtotal hysterectomy was performed (Figure 4). The postoperative period was successful and on the fourth day, the patient was discharged. The pathologic anatomy revealed rupture on the anterior surface of the uterus measuring 2.8 × 2.5 cm. The endometrial mucosa protruded through the solution of continuity. When the uterine cavity was cut, an endometrium with a thickness of 1.1 cm was observed.

Case Four: A 36-year-old presented at 12 weeks of gestation by her last menstrual cycle with vaginal bleeding and abdominal pain. The patient’s medical history was significant for one previous cesarean delivery four years prior. She was referred for ultrasound examination with a differential diagnosis of myoma or uterine malformation. Ultrasound findings showed a gestational sac in the anterior myometrium (in the area of the previous uterine scar) with an empty uterine cavity and cervical canal, along with thin or absent myometrium between the gestational sac and the bladder (Figure 5). When using the color Doppler, a marked peritrophoblastic flow was observed, typically useful in differentiating the bladder invasion, which is characterized by a low resistance, low impedance and high speed flow (Figure 6). Differential diagnosis with myoma was made as its vascularization was located in the periphery and had a high resistance flow. Given the diagnosis of CSEP, the patient was admitted; as the patient was stable, a laparotomy for resection of the trophoblastic tissue was performed (Figure 6). We examined the vesico-vaginal space and, approximately 2 to 3 cm away from the external cervical os, the location of the suspected ectopic pregnancy. The scar of the previous cesarean section was identified and the ectopic pregnancy was found, at which point a massive bleeding occurred due to the complete rupture of the scar. Therefore, a total hysterectomy was performed. (Figure 7). In this case, it was decided to perform a hysterectomy because of the wide area of rupture and the massive bleeding during surgery due to rupture of the scar uterine and arteriovenous vessels. The decision to conserve the uterus due to the patient’s hemodynamic instability has a higher risk.

The post-operative period was uneventful and the patient was discharged on day 6, with follow-up after 6 months showing no complications.

## 3. Discussion

Cesarean scar ectopic pregnancy (CSEP) is defined as a pregnancy localized over the scar of a previous C-section and that is completely surrounded by myometrium and fibrotic tissue [9].

### 3.1. Pathogenesis

Although the pathogenesis has not been delineated, the most accepted theory for CSEP is that impaired wound healing following previous trauma creates a myometrial defect and subsequent scar in which the blastocyst implants [10].

Two types of CSEP have been described: in type 1 (endogenic), the gestational sac grows inward toward the cervical isthmus space (with a potential for carrying to term); in type 2 (exogenic), the gestational sac grows outward toward the bladder and abdominal wall [11]. Determination of the type may help with counseling on expectant management or the medical/surgical approach for termination. Type 1 presentation may be milder and even asymptomatic during the first trimester of pregnancy because of its ability to grow into the uterine cavity.

More than 30 different treatment modalities for CSEP have been reported: the success rate and associated morbidity and mortality vary with each method and are dependent on patient stability and desire for future fertility.

Here, we presented four cases of CSEP, diagnosed via transvaginal and abdominal ultrasound, three of which required surgical treatment and only one of which was resolved with medical treatment. Case One appears to be CSEP type 1 with implantation on the area of the scar with progression to the cervical isthmus and uterine cavity. By contrast, for the other three cases, the gestational sac was implanted in the area of the scar, causing its rupture and bleeding during the first trimester of pregnancy.

### 3.2. Diagnosis of CSEP

Early diagnosis of CSEP is necessary to avoid the high risk of maternal bleeding and associated morbidity and mortality that can occur if uterine rupture occurs. The most common presentations include vaginal bleeding, generalized abdominal pain, and previous history of cesarean section.

Transvaginal ultrasound is the preferred test for diagnosis of CSP, with a sensitivity of 86.4% [12], leaving other diagnostic techniques, such as magnetic resonance, only for those cases in which a clear diagnosis is not obtained, there is a high suspicion, and the clinic allows it. For this reason, the majority of CSPs have been diagnosed by transvaginal scan in the early weeks of pregnancy. A sagittal view along the long axis of the uterus through the gestation sac can localize a CSP with confidence; furthermore, it permits the measurement of the thickness of the myometrium between the gestation sac and the bladder when less than 5 mm [12,13]. This thickness of myometrium has been shown in half of cases. By way of diagnosis, Jurkovic et al. have described a negative “sliding organ sign”, defined as the inability to displace the gestational sac from its position at the level of the internal os by gentle pressure applied by the transabdominal probe [13].

Additional diagnostic information can be obtained by color flow Doppler to show distinct circular peritrophoblastic perfusion surrounding the gestational sac that can help delineate the CSP sac with location of the placenta in relation to the scar and proximity to the bladder [13] (Table 1).

### 3.3. Treatment Options

Due to the rarity of the condition, a majority of CSPs are case reports or small case series reported in the literature, with no consensus on the preferred mode of treatment. Treatment should be tailored to the individual patient. Desire for future fertility, size and gestational age of the pregnancy, and hemodynamic stability should be considered when determining a treatment plan. Treatment objectives should be to perform embryo reduction prior to rupture, to remove the gestation sac, and to preserve the patient’s future fertility. Gestational age and viability, evidence of myometrial deficiency, and clinical symptoms at presentation have been considered by various authors to determine the management (Figure 8).

Several treatment options have been used employed; these can be categorized as medical or combined, uterine artery embolization (UAE), surgical and combination.

#### 3.3.1. Medical Treatment or Combined

Systemic administration of methotrexate (MTX) is a standard treatment for tubal ectopic pregnancy. There should be no reason to doubt its efficacy in CSEP. A dose of 50 mg/m^2^ or 1 mg/kg has proven to be useful. It has been shown that more than 50% of patients treated with medical treatment need a secondary procedure for successful treatment of CSEP. Medical treatment has therefore been combined with surgical aspiration of the sac, guided by ultrasound, in some cases [14]. In our first case, the patient had no desire for future fertility, and in view of the growth of the gestational sac into the cavity, medical treatment was chosen with a successful surgical ultrasound-guided aspiration.

This method has been successfully combined with local injection of MTX, potassium chloride [15], hyperosmolar glucose, and crystalline trichosanthin [10]. Under ultrasound guidance, MTX can be injected locally into the gestation sac via a transabdominal or transvaginal route. The transabdominal route requires a longer needle, used with caution not to penetrate the bladder wall, and does not require any anesthesia. The transvaginal approach allows for a shorter distance to the gestation sac with minimal risk of bladder injury.

Conservative medical treatment is suitable for an asymptomatic patient, <8 weeks of gestation, with β-hCG levels <5000, embryo with cardiac activity, and a myometrial thickness of <2 mm between the CSP and the bladder [15]. Medical treatment alone or in combination has the advantage of being less aggressive and preserving fertility, but requires time and patience. Its disadvantages are that it may take 4–16 weeks for β-hCG to drop to normal [16], the risk of rupture and hemorrhage, and the non-resolution of a possible alteration at the level of the C-section scar, which may eventually lead to recurrences or other complications such as placental accretism or increased risk of uterine rupture.

#### 3.3.2. Uterine Artery Embolization (UAE)

Uterine artery embolization (UAE) has been used to reduce the risk of subsequent hemorrhage in patients who undergo medical treatment or conservative surgery [17]. UAE pre-treatment was associated with significantly decreased blood loss and length of hospitalization. Patients who desire future fertility should be counseled regarding the risks of pregnancy after UAE: preterm labor, malpresentation, miscarriage, and postpartum hemorrhage. UAE is not considered a first-line option for patients who desire future fertility [18], due to its high failure rate, complication rates, and potential for a detrimental impact on future fertility.

#### 3.3.3. Surgical

Several additional techniques have been described as treatment of CSEP, including dilation and curettage; direct excision of CSEP via an abdominal, laparoscopic, or hysteroscopic approach; and definitive management with hysterectomy. Ultrasound-guided dilation and curettage are not recommended, as they do not provide a view of the cavity nor the exact location of the gestational sac. A high risk of uterine rupture and severe bleeding remains, which may force hysterectomy on a secondary basis [10]. The advantage of hysteroscopy and gestational sac removal and scar repair via laparoscopy is that they are less invasive, with less bleeding and less time spent in hospital. All these techniques are suitable for women who desire to preserve their fertility.

In patients without a desire for future fertility, as in our second case, hysterectomy is an appropriate technique to take into account.

## 4. Conclusions

Herein, we presented the rare pathology of CSEP that is becoming more and more frequent, for which transvaginal ultrasound and flow Doppler provide high diagnostic accuracy. Early diagnosis offers treatment options that can help avoid uterine rupture and bleeding, thus preserving the uterus and future fertility. Although there are no clear guidelines for treatment, we recommend individualized treatments for each patient with this pathology, depending on their personal characteristics. Total hysterectomy is likely the most appropriate treatment for those patients who do not desire future fertility.

## Figures and Tables

**Figure 1 medicina-57-00362-f001:**
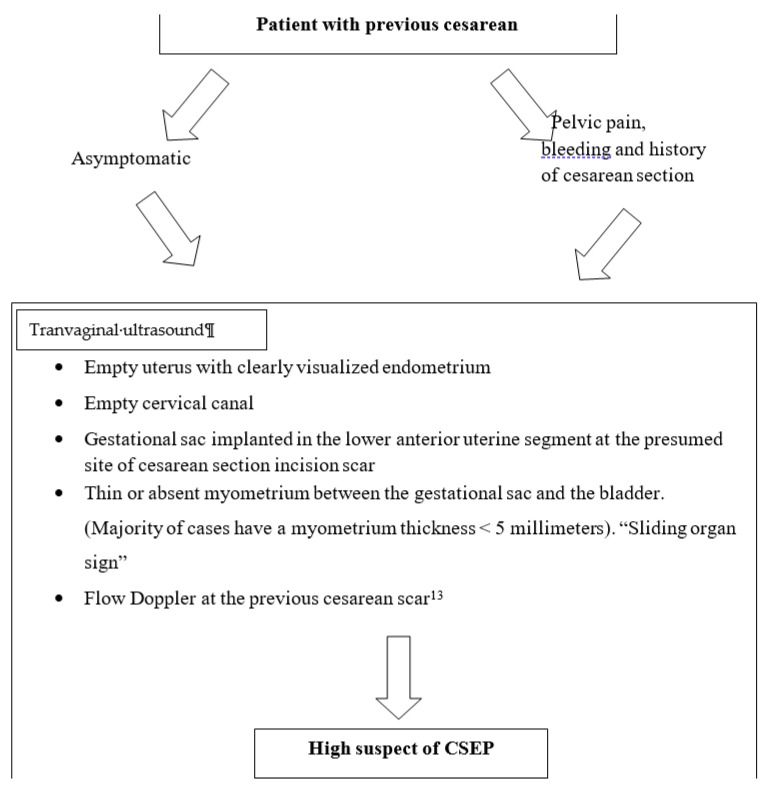
Cesarean scar ectopic pregnancy.

**Figure 2 medicina-57-00362-f002:**
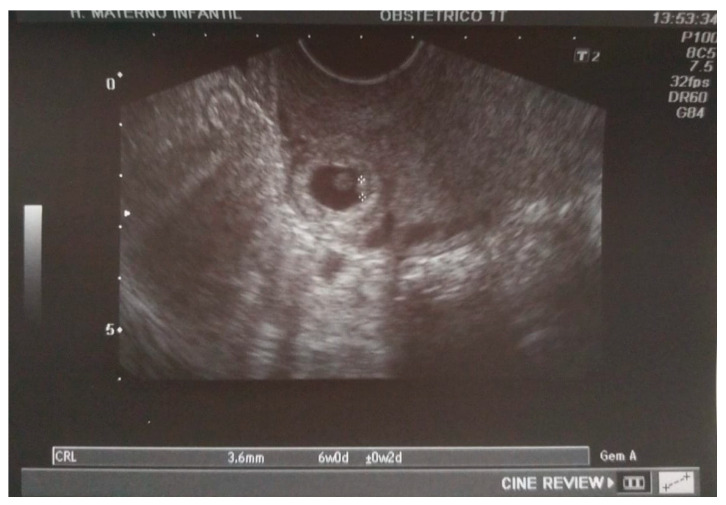
A sagittal transvaginal image showing a gestational sac implanted in the scar area of the previous cesarean with a 3.6 mm embryo.

**Figure 3 medicina-57-00362-f003:**
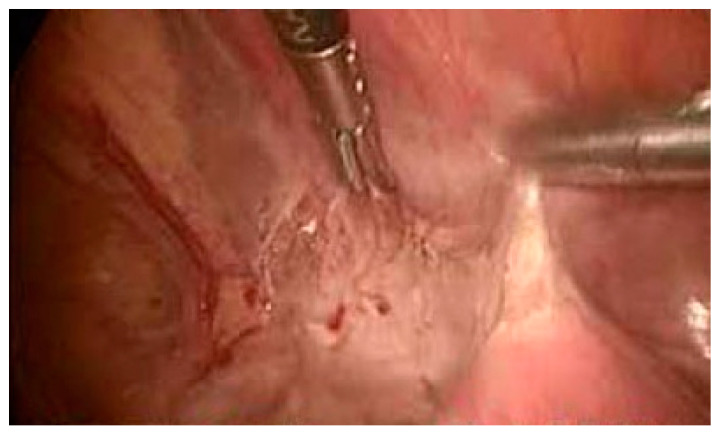
The bladder was released from a gestational sac.

**Figure 4 medicina-57-00362-f004:**
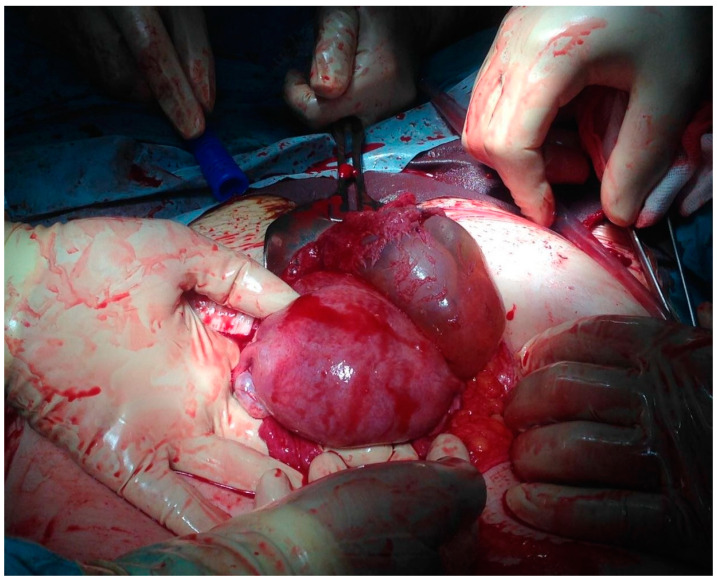
A 13-week visible gestational sac and prolapse after rupture of the uterine segment.

**Figure 5 medicina-57-00362-f005:**
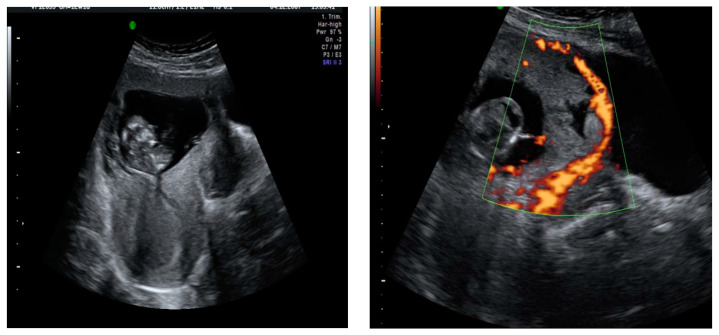
Ultrasound displaying (at **left** and **right**) a thinned myometrium area with a solution of continuity between the gestational sac and the bladder.

**Figure 6 medicina-57-00362-f006:**
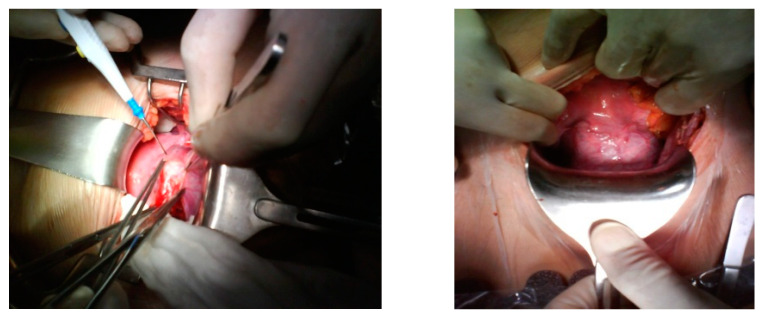
Gestational sac visible through myometrial anterior aspect.

**Figure 7 medicina-57-00362-f007:**
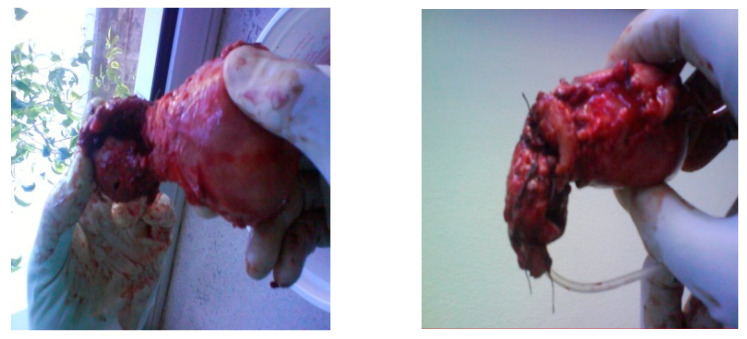
Section of hysterectomy with ectopic pregnancy.

**Figure 8 medicina-57-00362-f008:**
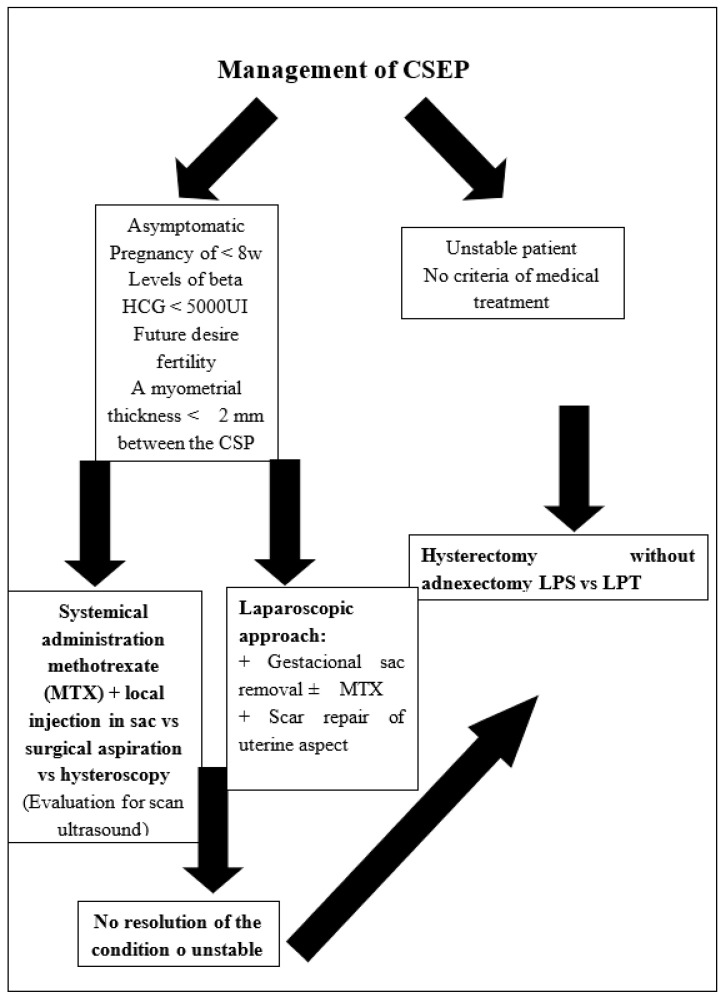
Management of CSEP; LPS: Laparoscopy, LPT: Laparotomy HCG: BetahCG.

**Table 1 medicina-57-00362-t001:** Diagnostic Criteria.

CSEP Diagnostic Criteria [13]
1. Empty uterus with clearly visualized endometrium
2. Empty cervical canal
3. Gestational sac implanted in the lower anterior uterine segment at the presumed site of cesarean section incision scar
4. Thin or absent myometrium between the gestational sac and the bladder. (Majority of cases have a myometrium thickness <5 mm). “Sliding organ sign”
5. Doppler flow at the previous cesarean scar [13].

The following ultrasound criteria have been put forward for the diagnosis of a CSP.

## Data Availability

The information related to the cases can be obtained from the corresponding author (Lorena Sabonet Morente) upon reasonable request. The data were collected during the patients’ hospital admission by the investigators who provided direct assistance. These data are stored in the computerized support system of electronic medical records of the Andalusian health system.

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
