# Peer review of "Cesarean Scar Ectopic Pregnancy—Case Series: Treatment Decision Algorithm and Success with Medical Treatment"

_medicina, 2021, doi:10.3390/medicina57040362_

Round 1
Reviewer 1 Report
A case series consisting of 4 patients about Cesarean Scar Ectopic Pregnancy with a small mini-review on current management of Cesarean Scar Ectopic Pregnancy. The condition is becoming more and more common, so case series, although interesting, may not have enough priority to be published, as the current incidence is estimated to be 1 in 1000/2000.
the only suggestion I have is to add the suffix: a case series to the title.
Thank You
Reviewer 2 Report
Thank you for the possibility to review your paper. My remarks are in the attached pdf file.

Round 2
Reviewer 1 Report
The authors addressed all comments. The paper is publishable.